# CysB Is a Key Regulator of the Antifungal Activity of *Burkholderia pyrrocinia* JK-SH007

**DOI:** 10.3390/ijms24098067

**Published:** 2023-04-29

**Authors:** Meng Yu, Yuwei Tang, Lanxiang Lu, Weiliang Kong, Jianren Ye

**Affiliations:** Co-Innovation Center for Sustainable Forestry in Southern China, College of Forestry, Nanjing Forestry University, Nanjing 210037, China; ym@njfu.edu.cn (M.Y.); m1_mang@163.com (Y.T.); llx@njfu.edu.cn (L.L.); k3170100077@njfu.edu.cn (W.K.)

**Keywords:** *Burkholderia pyrrocinia* JK-SH007, CysB, Cysteine, Fe–S clusters, ornibactin

## Abstract

*Burkholderia pyrrocinia* JK-SH007 can effectively control poplar canker caused by pathogenic fungi. Its antifungal mechanism remains to be explored. Here, we characterized the functional role of CysB in *B. pyrrocinia* JK-SH007. This protein was shown to be responsible for the synthesis of cysteine and the siderophore ornibactin, as well as the antifungal activity of *B. pyrrocinia* JK-SH007. We found that deletion of the *cysB* gene reduced the antifungal activity and production of the siderophore ornibactin in *B. pyrrocinia* JK-SH007. However, supplementation with cysteine largely restored these two abilities in the mutant. Further global transcriptome analysis demonstrated that the amino acid metabolic pathway was significantly affected and that some sRNAs were significantly upregulated and targeted the iron–sulfur metabolic pathway by TargetRNA2 prediction. Therefore, we suggest that, in *B. pyrrocinia* JK-SH007, CysB can regulate the expression of genes related to Fe–S clusters in the iron–sulfur metabolic pathway to affect the antifungal activity of *B. pyrrocinia* JK-SH007. These findings provide new insights into the various biological functions regulated by CysB in *B. pyrrocinia* JK-SH007 and the relationship between iron–sulfur metabolic pathways and fungal inhibitory substances. Additionally, they lay the foundation for further investigation of the main antagonistic substances of *B. pyrrocinia* JK-SH007.

## 1. Introduction

The *Burkholderia cepacia* complex (Bcc) is a group of closely related Gram-negative bacteria with a continuously evolving taxonomy and is widely distributed across terrestrial and aquatic ecosystems [1]. These bacteria interact differently with their hosts and may establish beneficial, neutral, or detrimental associations with their hosts depending on their lifestyle and pathogenicity. Pathogenic *Burkholderia* species commonly cause pulmonary infection and pose a great danger to people with cystic fibrosis (CF), as infection by these species may result in increased patient morbidity and mortality [2,3,4,5,6]. Without doubt, beneficial strains play different roles. A few bacteria belonging to the *Burkholderia* genus are commonly present in the rhizosphere and endosphere of plants and are able to enhance plant growth [7,8,9,10]. For example, some Bcc strains regulate iron uptake through the ferric uptake regulator (Fur) in an iron concentration-dependent manner to produce siderophore [11]. Other strains can produce some secondary metabolites (including siderophore and lipopeptides) in a non-ribosome manner through a non-ribosome peptide synthase called NRPS [12]. Other strains are able to produce the secondary metabolite pyrrolnitrin through tryptophan metabolism [12,13].There also exist strains capable of synthesizing antimicrobial substances, such as cepain, via quorum sensing (QS) [14,15,16,17]. As a beneficial Bcc strain, *B. pyrrocinia* JK-SH007 has attracted attention for its superior ability to control poplar canker. Poplar canker is a destructive disease of poplar trunk in poplar production and afforestation. This disease has the characteristics of rapid development, wide distribution, and serious harmful consequences, seriously affecting the yield, quality, and afforestation effect of poplar trees [18]. *B. pyrrocinia* JK-SH007 has been reported to exert a significant antagonistic effect against *Phomopsis macrospora*, one of the pathogenic fungi causing poplar canker [19]. However, while the importance of the antifungal process of *B. pyrrocinia* JK-SH007 has been widely studied, the molecular regulatory mechanisms of genes related to its antagonistic function are still unknown, and the relationship between its antimicrobial substances and metabolic pathways needs to be explored.

Fe–S clusters play an essential role in the life activities of bacteria and can influence biological life processes such as respiration, electron transfer, amino acid metabolism, and DNA repair [20,21,22,23]. However, Fe–S cluster biosynthesis requires the availability of sulfur from the cysteine pool. Cysteine is the most critical sulfur-containing compound involved in microbial sulfur metabolism, and the cysteine synthesis pathway is central to microbial sulfate assimilation, influencing the conversion of inorganic sulfur to organic sulfur [24]. In many bacteria, Fe–S clusters can be involved in processes related to virulence [25,26]. For example, SufBCD proteins have been shown to influence the virulence of *Erwinia chrysanthemi* by affecting the assembly of Fe–S clusters under oxidative stress and iron limitation [27]. In carriers of Fe–S clusters, deletion of the Nfu gene negatively affects virulence, as observed in *Staphylococcus aureus* [28]. Although the effect of Fe–S clusters on pathogenic bacterial virulence has been demonstrated in many reports, the effect of Fe–S clusters on *B. pyrrocinia* JK-SH007 virulence is still unknown.

Transcriptional regulators are important molecules that control gene expression and directly affect the degree of gene expression to regulate related pathways. CysB is a key transcriptional regulator involved in cysteine synthesis and can influence the expression of many cys genes, which in turn regulate sulfur metabolic pathways in bacteria, including thiosulfate and sulfate transport and sulfate reduction [29,30,31]. In *Salmonella typhimurium*, CysJIH can affect superoxide dismutase activity and the production of reactive oxygen species, reduced thiols, and H_2_S, all of which are associated with bacterial self-protection mechanisms, and CysB is a positive regulator of CysJIH [32]. In addition, CysB is able to positively regulate the expression of CysU and CysI, which have been shown to play roles in sulfate and thiosulfate transport and sulfate reduction and are essential for cysteine synthesis [33]. Therefore, the question of whether the CysB participates in the antagonism of *B. pyrrocinia* JK-SH007 against poplar canker pathogens and how to regulate its antifungal activity remains to be further explored. Here, we obtained a mutant strain that lacked antagonism against *P. macrospora* by screening a mutant library of *B. pyrrocinia* JK-SH007. The mutated gene was identified as *cysB* in the target transposon mutant. Additionally, we try to explain whether there is a certain relationship between the antifungal activity of *B. pyrrocinia* JK-SH007 and the related pathways of CysB regulation in the Bcc family in order to further reveal the molecular antagonistic mechanism of *B. pyrrocinia* JK-SH007.

## 2. Results

### 2.1. Differences in the Activities of WT and Mutant Strains of B. pyrrocinia JK-SH007 against Phomopsis Macrospora

The effect of the wild type (WT) compared with that of the mutant strains on the growth of *P. macrospora* was tested on PDA plates. Then, the mutant strain was co-inoculated with *P. macrospora* on PDA plate medium for plate antagonism experiments. The size of the antagonistic inhibition zone of the WT was used as a reference, and the mutant V49 with significantly reduced antagonistic function was identified by screening (Figure 1A). The one-step growth curves of the WT and mutant V49 were measured separately. The results demonstrated that the growth rates of the two strains during the exponential growth period were almost identical, and the growth rate of the mutant V49 was slightly higher than that of the WT after reaching the stable phase. Moreover, the growth of the mutant V49 showed a decreasing trend after 60 h of incubation (Figure 1D). We further determined the inhibition rates of the WT and mutant V49, and the results showed a significant difference in inhibition between the WT and mutant V49. The inhibition rate of the WT against *P. macrospora* was 42.22% compared to that in the control group treated with sterile water, and the inhibition rate of the mutant V49 was reduced by approximately 20% compared to that of the WT (Figure 1B,C).

### 2.2. Localization and Southern Blotting of Mutated Genes in the Mutant V49

The genomic DNAs of the WT and mutant V49 were digested with the restriction endonuclease *Eco*R I (without cleavage within Tn5 and Tmp) and probed by Southern blotting with DNA fragments containing Tmp. The Southern hybridization results showed that Tmp was present in the genomic DNA of the mutant V49 and was not present in the genomic DNA of the WT (Figure 2A). Southern blotting showed that the hybridizable fragment generated by *Eco*R I digestion was approximately 1200 bp in length, and the results indicated that the reduced antagonistic ability of the mutant V49 could be attributed to a single gene mutation. The complete sequence of the DNA on both sides of the transposon was determined by reverse PCR amplification and further compared with genomic data to identify the gene with the transposon insertion as *cysB* (full length, 942 bp) (Figure 2B). In conclusion, the results suggest that the LysR family transcriptional regulator CysB is essential for the antifungal activity of the WT.

### 2.3. Differences in Cysteine Content

To further investigate the effect of CysB on cysteine synthesis, we measured the cysteine content of the WT and mutant V49. The results showed that the extracellular cysteine content of both the WT and mutant V49 was greater than the intracellular content, and the overall cysteine content of the WT was higher than that of the mutant V49 (Figure 3A). The extracellular cysteine levels of the mutant V49 strain incubated with 50 μmol/L cysteine for different durations within 24 h were measured separately. The results showed that, after 8 h of incubation, the extracellular cysteine content of both MSA + V49 + cysteine and MSA + V49 was higher than that of the control group. Moreover, the extracellular cysteine content of both experimental groups gradually decreased with time and was finally slightly lower than the cysteine content in the culture medium and stabilized at a certain dynamic level (Figure 3B).

### 2.4. Effect of Adding L-cysteine on the Antagonistic Ability of the Mutant V49

L-cysteine at concentrations of 50 μmol/L, 100 μmol/L, 200 μmol/L, 300 μmol/L, and 400 μmol/L was added to the PDA plates, and the results demonstrated that the antifungal activity of the mutant V49 was gradually restored as the concentration of added L-cysteine increased. When the concentration of added L-cysteine reached 100 μmol/L, the antifungal activity of the mutant V49 was almost completely restored to WT levels, and when the concentration of added L-cysteine reached 200 μmol/L, the phenotype of the mutant V49 also returned to yellow (Figure 4A). The difference in the inhibition rate was also measured, and when the concentration of added L-cysteine reached 100 μmol/L, the inhibition ability of the mutant V49 was almost indistinguishable from that of the WT (Figure 4B).

### 2.5. HPLC–MS Analysis to Identify Differentially Abundant Substances

To identify the inhibitory substance whose synthesis was affected by cysteine, HPLC–MS/MS was further used to target the relevant substances. The mutant V49 lacked two peaks, at 3′08″ and 4′41″, in comparison to the WT. After the addition of cysteine, the LC–MS results showed two peaks, at 3′08″ and 4′41″, and the peak areas were almost identical (Figure 5). Further analysis of these two peaks showed a parent ion with a molecular weight of 709.3743 in the outgoing peak at 3′08″, and a parent ion with a molecular weight of 737.4023 was found in the outgoing peak at 4′41″, which is consistent with the mass-to-charge ratio of the parent ions of siderophores. The samples were subjected to secondary fragmentation to identify the fragment ions and compare the data against the PubChem database. The emergent peak at 3′08″ was consistent with the daughter ion of ornibactin C6 in PubChem, and the emergent peak at 4′41″ was consistent with the daughter ion of ornibactin C8 in PubChem (Appendix A). It was determined that the disruption of CysB affected the synthesis of the siderophore ornibactin in WT, and the addition of cysteine restored the production of the siderophore ornibactin in WT.

### 2.6. The Addition of L-cysteine Restored the Siderophore Production Ability of the Mutant V49

Siderophore production by the WT and mutant V49 was analyzed based on the HPLC–MS/MS results, and the CAS plate results demonstrated that the ability of the mutant V49 to produce siderophores was significantly reduced compared to that of the WT (Figure 6B). To further verify whether the addition of L-cysteine would restore the ability of the mutant V49 to produce siderophores, the mutant V49 strain was cultured on MSA medium supplemented with 100 μmol/L, 200 μmol/L, 300 μmol/L, 400 μmol/L, 500 μmol/L, 600 μmol/L, or 700 μmol/L cysteine. The results demonstrated that the addition of different concentrations of cysteine restored the siderophore production ability of the mutant V49 to different degrees. Moreover, when the added L-cysteine concentration reached 700 μmol/L, there was almost no difference in siderophore production ability between the mutant V49 and WT (Figure 6A,C). This demonstrated that L-cysteine was associated with the synthesis of siderophores in the WT.

### 2.7. The Transcriptome of B. pyrrocinia JK-SH007 Was Affected When CysB Was Disrupted

The total RNA of the WT and mutant V49 was extracted for Illumina HiSeq sequencing. The analysis showed that all the sequencing data were of good quality (Q30 = 91.42%) (Appendix A). The number of upregulated genes (*n* = 749) was more than six times higher than that of downregulated genes (*n* = 132) (Figure 7A). In addition, we performed KEGG enrichment analysis of the differentially expressed genes (DEGs) (Figure 7B). The results demonstrated that the upregulated genes were mainly concentrated in the ABC transport, pyruvate metabolism, amino acid metabolism, and fatty acid metabolism pathways, and the downregulated genes were mainly concentrated in the amino acid metabolism and signal transduction pathways. To check the reliability of the transcriptomic results, qPCR was performed on the genes of interest, and the results were consistent with the transcriptomic trends (Appendix A). We analyzed the key downregulated genes and found that disruption of CysB affected the expression of genes related to iron–sulfur metabolism; in addition, some small RNAs (sRNAs) were found to be involved in the regulation. We selected eight genes of interest for qPCR to verify that the expression or upregulation of these genes was associated with iron–sulfur cluster-mediated bacterial virulence processes [34,35,36,37] and that the expression of these genes was decreased in the mutant V49 (Figure 8).

### 2.8. Predicted Targets of sRNAs

To further explore the role of sRNAs identified in the transcriptome, we calculated and predicted the possible mRNA targets of Bp_007_sr1, 2, 3, and 4 using the default parameters of TargetRNA2. This computational search was performed considering all annotated open reading frames (orf) in *B. vietnamiensis* G4, *B. thailandensis E264, B. phenoliruptrix* BR3459a, *B. cenocepacia* J2315, *B.* sp. CCGE1001, *B.* sp. CCGE1002, *B. phytofirmans* PsJN, *B. pseudomallei* NCTC 13179, *B.* sp. KJ006, and *B.* sp. 383. TargetRNA2 was used to predict the possible sites of action of these sRNAs. Here, we list only the targets predicted to be relevant to the process of iron–sulfur metabolism (Table 1). By prediction, we found that Bp_007_sr1, 2, and 3 mainly target the iron–sulfur cluster assembly process, and the putative targets are mainly 2Fe-2S iron–sulfur cluster proteins and Fur proteins, which are important players in maintaining iron homeostasis [38,39]. In addition, Bp_007_sr4 is upregulated after CysB is disrupted, and it mainly targets cysteine desulfurase, which is required for Fe–S cluster biosynthesis [40].

## 3. Discussion

*Burkholderia cepacia* has been used in recent years as an effective biocontrol agent to replace traditional pesticides in the management of plant diseases, and it has multiple mechanisms related to the suppression of plant disease development [41,42,43,44]. However, the antagonistic mechanisms by which some *B. cepacia* strains inhibit the growth of pathogenic microorganisms have not been fully resolved. Here, we screened a mutant that affected the growth of the poplar canker pathogen through transposon mutagenesis and found that the insertional mutation was located in the transcriptional regulator of sulfur metabolism, CysB. We demonstrated that CysB affected the biosynthesis of cysteine in *B. pyrrocinia* and fully restored the inhibitory effect of the mutant strain on the pathogenic fungus *P. macrospora* by cysteine supplementation. When the extracellular cysteine reached a certain level, it was taken up by the mutant V49 through a transmembrane pathway, which indicated that the intracellular transport of cysteine was not affected by CysB. In addition, CysB was able to regulate the expression of CysD and CysE, which play a role in sulfate assimilation and cysteine synthase complex synthesis, respectively, and are essential for cysteine synthesis [45,46]. Thus, as a LysR-type transcriptional regulator, CysB usually acts as a transcriptional activator of cys genes to facilitate sulfate assimilation, sulfate-thiosulfate transport, sulfate reduction, and cysteine synthesis [47,48]. Moreover, CysB is able to positively regulate the expression of type III secretion system (T3SS) genes in vitro and in plants through the PrhG to HrpB pathway, which is involved in the virulence process of *Ralstonia solanacearum* [33]. This demonstrates that CysB can be indirectly involved in the virulence process of some bacteria.

In addition, the destruction of CysB resulted in a decrease in the amount of ornibactin synthesized. Ornibactin is a Bcc-specific siderophore and is the most common type of siderophore, providing hexadentate ligands in 1:1 equivalents to form stable complexes with ferric ions. The activity of ornibactin is probably highest among types of siderophores that can be synthesized by Bcc, but the exact stability constants have not been reported [38,39]. The synthesis of both ornibactin C6 and ornibactin C8 was significantly restored by cysteine supplementation, and we speculate that this may be correlated with the recovery of its antagonism against pathogenic fungi, as ornibactin is thought to be involved in the production of some antifungal substances [49]. Unfortunately, we have not been able to obtain pure ornibactin, and it is unknown whether this compound has fungal inhibitory activity. Furthermore, the addition of cysteine did not have a significant effect on bacterial growth, which demonstrated that the recovery of ornibactin was not caused by changes in the growth of the mutant V49 but rather by the entry of extracellular cysteine into the cell via transmembrane transport to participate in the synthesis of siderophores (Figure 1D). Therefore, we suggest that cysteine is a key substance involved in the production of the siderophore ornibactin by *B. pyrrocinia* and a key factor affecting virulence.

The results regarding cysteine and siderophores all seem to point to an important biological process in bacteria: the biosynthesis of Fe–S clusters. Cysteine plays the role of an S supplier in this process, while siderophores are responsible for Fe delivery to the bacteria [48,50,51]. This pathway seems to be blocked due to the destruction of CysB. Interestingly, we identified some significantly upregulated sRNAs in the transcriptome. The prediction and validation of these sRNA binding sites combined with qPCR indicate that they may affect some key processes in the biosynthesis of Fe–S clusters. These processes involve the synthesis of iron sulfur cluster assembly proteins, cysteine desulfurization enzymes, iron-binding protein IscA, iron uptake regulatory protein Fur, and iron sulfur binding oxidoreductases (Table 2). Since sRNA inhibits translation initiation through ribosomal occlusion mainly through base pairing, increased sRNA levels inhibit the normal transcription control of genes involved in the corresponding biological process [52]. In *Escherichia coli*, the sRNA RyhB is able to promote siderophore production and coordinate Fe–S cluster cofactor biogenesis by inhibiting depleted iron utilization proteins [53]. Thus, the biological processes affected by sRNA are necessary for the biosynthesis of Fe–S clusters, and changes in these processes may lead to the destruction of Fe–S clusters, as perhaps evidenced by the decrease in the ornibactin level leading to a reduced need for iron for Fe–S cluster assembly. Regarding this process, we hypothesize that CysB affects both cysteine synthesis and some sRNA regulatory processes in the mutant V49. On the one hand, the decrease in cysteine synthesis leads to limited availability of S for Fe–S synthesis [53], blocking intracellular Fe–S synthesis, and the Fe^2+^ that fails to bind to S accumulates, leading to an increase in the intracellular Fe^2+^ concentration (Figure 9). Reduced cysteine synthesis affects the synthesis of the C-terminal cysteine cluster of OrbS, causing intracellular Fe^2+^ to impair the binding of OrbS to core RNAP (a multisubunit enzyme with a core catalytic structure), thereby inhibiting OrbS-dependent ornibactin production [54]. On the other hand, sRNA appears to affect the processes involved in Fe–S cluster synthesis, including cysteine desulfurization, Fe uptake, and Fe–S cluster assembly. The superposition of the dual effects resulted in a significant decrease in the amount of S bound to Fe, affecting the total amount of Fe–S clusters. It has been shown that Fe–S clusters are able to assimilate Fe^3+^ carried by siderophores dependent on the Suf pathway into Fe^2+^ for utilization and that siderophores also respond to intracellular Fe^2+^ concentrations [27]. Additionally, Fe–S can affect the activity of Fe/S protease toward cytochromes, which seems to explain the color changes that occur in bacteria after CysB is disrupted [55,56].

Fe–S clusters perform important biological functions, including electron transfer [57,58], substrate binding and activation [59,60], Fe/S storage [56,61], and regulation of enzyme activity [62]. Many Fe–S proteins (genes) associated with virulence have been identified [63,64]. For example, in *Pseudomonas aeruginosa*, the mutation of the scaffold protein NfuA, which is necessary for the maturation of the protein Fe–S center, will make the bacteria sensitive to fluoroquinolones under aerobic conditions. Therefore, the NfuA protein will maintain the growth and toxicity of *P. aeruginosa* under different stress conditions (oxidative stress, anaerobic, iron deficiency) [65]. There are also studies indicating that the deletion of the key gene Nfu for the maturation of Fe–S protein in *Staphylococcus aureus* can have a negative impact on the physiology and pathogenicity of *S. aureus* and can also lead to DNA damage [63]. We demonstrated that CysB deficiency impedes cysteine and Fe–S cluster biosynthesis, which may explain the reduced ability of *B. pyrrocinia* JK-SH007 to antagonize pathogenic fungi. Moreover, cysB has been shown to affect the T3SS and pathogenicity of *R. solanacearum*, which may be partly caused by the absence of the Fe–S cluster [66], as IscR in the Fe–S cluster of *Yersinia* has been identified as a regulator of the T3SS and a key factor involved in its pathogenesis [67]. Clearly, proteins containing iron–sulfur clusters are important for the expression or activity of the bacterial T3SS. We speculate that the lack of antagonistic ability of *B. pyrrocinia* JK-SH007 is due to the disruption of the biological function of the Fe–S cluster and the consequent loss of activity of some virulence factors. The addition of cysteine reactivated this pathway of the Fe–S cluster, which corresponds to the restoration of siderophore synthesis and antagonism, but the exact components of the Fe–S cluster of *B. pyrrocinia* JK-SH007 and the mechanisms that influence the production of virulence factors need to be further explored.

Taken together, the novel results of this study indicate that CysB is a regulator of the iron–sulfur metabolic pathway in *B. pyrrocinia*. Figure 9 shows our predicted working model of how CysB in *B. pyrrocinia* may affect iron–sulfur cluster synthesis, helping to further reveal the regulatory pathways of iron–sulfur metabolism within the cell and further explore the role of iron–sulfur clusters in the antimicrobial capacity of bacteria. Although we failed to demonstrate a clear antagonistic mechanism, our results contribute significantly to the understanding of the complex regulatory system of iron–sulfur metabolism in bacteria. Overall, our study reveals the important role of the CysB gene in the Fe–S cluster biosynthetic pathway while suggesting a possible link between these processes and the ability to suppress the fungus in the plant probiotic *B. pyrrocinia*. All of these results provide new insights into the various biological functions of the transcriptional regulator CysB and its complex regulation of virulence factors in *B. pyrrocinia*.

## 4. Materials and Methods

### 4.1. Experimental Materials

*Burkholderia pyrrocinia* JK-SH007 (strain number: CCTCC M209028), its mutant strains, and *P. macrospora* (strain number: CCTCC AF2015001) were provided by the Jiangsu Key Laboratory for Prevention and Management of Invasive Species in Nanjing, China.

### 4.2. Mutant Screening

The mutant strains were cultured using Luria–Bertani medium supplemented with 1‰ Trimethoprim (Tmp) antibiotic. *Phomopsis macrospora* was inoculated in the center of a plate containing potato dextrose agar (PDA) medium. Then, 2 µL of the mutant bacterial suspension was inoculated at a distance of 2.0 cm from the fungus. The mutant strains were cultured in LB medium with the following formulation in 1 L of ultrapure water: tryptone, 10.0 g; yeast extract, 5.0 g; sodium chloride, 10.0 g; pH 7.2–7.4. The mutant strains were incubated in a constant-temperature biochemical incubator at 28 °C in an inverted position, and the difference between the inhibition zone diameters of the mutant strains and the *B. pyrrocinia* JK-SH007 WT strain was measured and recorded after 72 h. A plate culture was inoculated with four strains, with *B. pyrrocinia* JK-SH007 in the upper left corner; the remaining three strains were mutant strains. Mutants with reduced antagonistic ability were selected for antagonistic ability rescreening. The mutants with reduced antagonistic ability were then selected for antagonistic ability rescreening, for which a plate was inoculated WT *B. pyrrocinia* and a mutant strain. Screening of the existing mutation library (approximately 5800 strains) in *B. pyrrocinia* JK-SH007 showed that only V49 showed a stable decrease in antibacterial activity during the screening process. Further research will be conducted with V49 as the experimental object.

### 4.3. Antagonism Test

The antifungal activities of the WT and mutant V49 were determined separately using sterile water as the control group, and three replicate experiments were set up for each group. The diameter of the inhibition zone of the antagonistic strain in the PDA plate was measured and recorded, and the rate of mycelial growth inhibition was calculated as follows: relative inhibition rate = diameter of treated colonies/(diameter of control colonies − diameter of fungal cake) × 100%.

### 4.4. Localization of Mutant Genes

Genomic DNA from the WT and mutant V49 was extracted using the freeze–thaw cycle method [68]. The genomic DNA was monoenzymatically cleaved with the restriction endonuclease *Eco*R I. The monoenzymatic cleavage products were ligated using T4 ligase, and Tn5 reverse primers (Table 2) were designed to reverse amplify the autocyclized products. The amplification products were recovered and sequenced. The sequencing results were integrated with the Tn5 transposon sequence and the WT genomic DNA sequence to locate the gene with a Tn5 insertion.

### 4.5. Southern Blotting

Southern blotting was performed using the Digoxigenin (DIG) High Prime DNA Labeling and Detection Starter Kit I (Roche Applied Science, Penzberg, Germany). Tmp fragments were amplified by PCR and purified using the High Pure PCR Product Purification Kit (Roche Applied Science). The purified Tmp fragments were labeled using the DIG-High Prime solution from the kit.

### 4.6. Transcriptome Sequencing

WT and V49 were incubated at 28 °C and 200 rpm for 10 h, and the organisms were collected by centrifugation at 6000 rpm for 5 min. Among them, JK_SH007_1, JK_SH007_2, and JK_SH007_3 were the control groups, and V49_1, V49_2, and V49_3 were the experimental groups in which the CysB gene was mutated.

Total RNA was extracted using the mirVana miRNA Isolation Kit (Ambion, Austin, TX, USA) following the manufacturer’s protocol. Then, cDNA libraries were constructed. The quality of the constructed libraries was checked with an Agilent 2100 Bioanalyzer. Then, the libraries were sequenced using an Illumina sequencer, and 150 bp/125 bp paired-end reads were generated. cDNA library construction and Illumina sequencing were performed by OE Biotech Co., Ltd. (Shanghai, China). We used the known genome sequence of *B. pyrrocinia* JK_SH007 (NCBI accession: GCA_022809715.1) as a reference, and the expression abundance of each gene in each sample was identified by sequence similarity comparison. The expression levels of each transcript were calculated using the fragments per kilobase of exon per million reads mapped (FPKM) method in Rockhopper software [69]. Genes with a false discovery rate (FDR) of ≤0.05 and a fold change of ≥2 were regarded as DEGs. Then, GO and KEGG enrichment analyses of the DEGs were performed by hypergeometric distribution tests to determine the biological functions or pathways that were mainly affected by the DEGs.

### 4.7. Target Prediction for sRNAs

To predict the mRNA targets of novel sRNAs of *B. pyrrocinia* JK-SH007, online algorithms such as TargetRNA2 (http://cs.wellesley.edu/~btjaden/TargetRNA2/, accessed on 18 April 2023) [70] were used with default parameters.

### 4.8. Real-Time Quantitative PCR

The TRIzol method was used to extract total RNA from samples of the WT and mutant strains that showed antagonism on the plate in a 3-day experiment [71]. Total RNA was reverse-transcribed to cDNA using an Evo M-MLV RT Mix Kit with gDNA Clean for qPCR (AG11728, Accurate Biotechnology, Hunan, Co., Ltd., Hunan, China,), and a SYBR Green Premix Pro Taq HS qPCR Kit (AG11701, Accurate Biotechnology, Hunan, Co., Ltd.) was used to perform quantitative PCR (qPCR) validation. All qPCR primers were designed using the NCBI Primer design tool (https://www.nih.gov/, accessed on 18 April 2023) online (Table 1). The data obtained were analyzed using the 2^−ΔΔCt^ method to calculate the transcript levels of genes related to the iron–sulfur metabolic pathway at different incubation times [72].

### 4.9. L-cysteine Measurement

WT and V49 strains were cultured with modified sucrose-asparagine (MSA) at 28 °C and 200 rpm for 12 h. Each culture was then adjusted to 1 × 10^8^ cells ml^−1^. The difference in intracellular and extracellular cysteine content between the WT and V49 strains was determined using the cysteine (Cys) content assay kit (Beijing Solaibao Biotechnology Co., Ltd., Beijing, China). Then, the extracellular culture fluid and the milled cell filtrate samples were treated with cysteine extraction buffer. The determination of extracellular cysteine was achieved as follows: The experiment was conducted using LB culture medium as the control group and WT and V49 fermentation broth as the experimental group. The cysteine extract was mixed with blank LB medium, WT fermentation broth, and V49 fermentation broth in a 1:1 ratio to prepare the sample. The determination of intracellular cysteine was achieved through the following: the strains of WT and V49 were cultured at 28 °C for 14 h and collected by low-speed centrifugation. The bacterial body was washed twice with phosphate buffer, then suspended with 1.5 mL phosphate buffer before an ultrasonic crusher was used to break the bacterial body. Using phosphate buffer as the control group, samples were prepared by mixing cysteine extract with phosphate buffer, WT bacterial suspension, and V49 bacterial suspension in a 1:1 ratio. The samples were allowed to stand for 15 min at room temperature at a ratio of 1:4 with the detection reagent, and the absorbance values were measured at 600 nm. Then, the cysteine content was calculated. The extracellular cysteine content of V49 was determined in the same way at different time points (8, 12, 16, 20 and 24 h). Each treatment had three replicates, and the experiment was repeated three times.

### 4.10. CAS Assays to Detect Siderophore Production

Differences in the siderophore production ability of the WT and mutant V49 were examined using chrome azurol S (CAS) agar plate medium [73]. Single colonies of bacteria were incubated in MSA medium at 28 °C for 3 days. The amount of siderophores produced by the bacteria in MSA broth was determined using the CAS colorimetric method as described previously [74]. Briefly, bacterial cultures in MSA medium were centrifuged at 12,000 rpm for 10 min to obtain a cell-free supernatant. The supernatant was then incubated with CAS detection reagent in a 1:1 ratio for 1 h at room temperature, and the absorbance at 630 nm was measured. The absorbance of sterile MSA medium incubated with CAS solution was used as a reference. Each treatment had three replicates, and the experiment was repeated three times.

### 4.11. L-cysteine Addition Experiment

The growth curves of the WT and mutant V49 in media supplemented with different concentrations of L-cysteine were measured. Similarly, different concentrations of L-cysteine were added to PDA plate medium to observe the change in the antagonistic ability of the WT and mutant V49 against the pathogen. Moreover, different concentrations of L-cysteine were added to MSA liquid medium to culture the WT and mutant V49, and siderophore production was quantified with CAS assay solution.

### 4.12. HPLC–MS Analysis

HPLC–MS analysis of purified compounds was performed with an Agilent 1290 HPLC in tandem with an AB5600TOF mass spectrometer and examined using a C18 column (Waters Xbridge C18 2.1 × 150 mm, 3.5 μm). HPLC results were monitored by a UV absorption detector at 520 nm. The HPLC mobile phase, applied at a flow rate of 1 mL/min, was an acetonitrile/water gradient used to separate fractions over 40 min as follows: 0–9 min, phase B increased from 5% to 99.5%; 9–12 min, phase B 99.5%; 12–12.3 min, phase B from 99.5% to 5%; 12.3–15 min, phase B 5% to 60% to 95%. Phase A was water containing 0.1% formic acid, and phase B was 100% acetonitrile. The raw LC–MS data were then analyzed using PeakView 1.2 software and the PubChem website. The experiment was repeated twice.

### 4.13. Statistical Analysis

Analysis of variance (ANOVA) was used to determine the significance of the differences between the means of three or more groups.

### 4.14. Data Availability

Raw transcriptome data of the six samples were deposited in the SRA database (Sequence Read Archive, NCBI) with accession numbers SRR22411507, SRR22411508, SRR22411509, SRR22411504, SRR22411505, and SRR22411506.

## 5. Conclusions

Taken together, the novel results of this study indicate that CysB is a regulator of the iron–sulfur metabolic pathway in *B. pyrrocinia*. Figure 9 shows our predicted working model of how CysB affects the synthesis of iron–sulfur clusters in *Pyrrocinia*. This will help to further reveal the regulatory pathways of intracellular iron and sulfur metabolism and explore the effect iron and sulfur clusters have on bacterial antagonistic ability. Although we failed to demonstrate a clear antagonistic mechanism, our results contribute significantly to the understanding of the complex regulatory system of iron–sulfur metabolism in bacteria. Overall, our study reveals the important role of the CysB gene in the Fe–S cluster biosynthetic pathway while suggesting a possible link between these processes and the ability to suppress the fungus in the plant probiotic *B. pyrrocinia*. All of these results provide new insights into the various biological functions of the transcriptional regulator CysB and its complex regulation of virulence factors in *B. pyrrocinia*. At the same time, this also provides direction for our future research as we hope to explore the relationship between iron–sulfur clusters and antifungal substances by identifying key factors that maintain the stable state of Fe–S and tracking the trends of Fe and S elements to provide a theoretical basis for further identifying relevant antibacterial substances and developing biological control agents.

## Figures and Tables

**Figure 1 ijms-24-08067-f001:**
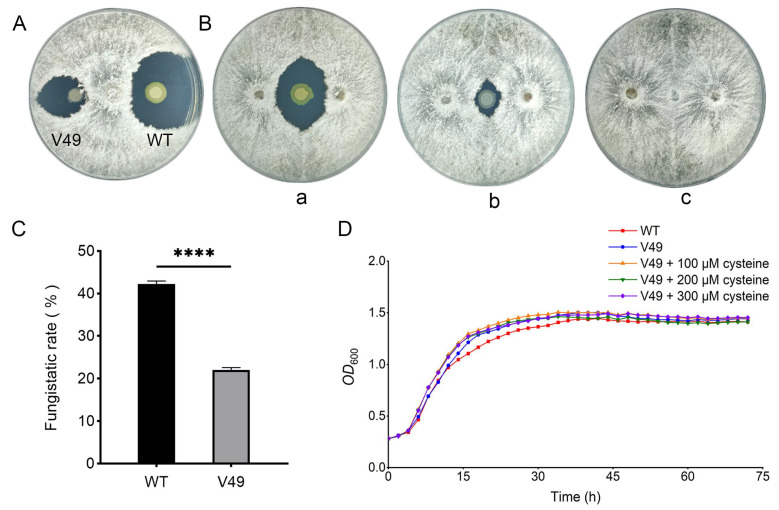
Determination of the fungal inhibition rate and growth curve. (**A**) Differences in antifungal activity between the WT and mutant V49. (**B**) Plate antagonism test: (a) WT; (b) mutant V49; (c) H_2_O. (**C**) Statistical analysis of the fungal inhibition rate. (**D**) Growth curves of strains with different concentrations of cysteine added to LB (Luria–Bertani) medium. ****, *p* < 0.0001.

**Figure 2 ijms-24-08067-f002:**
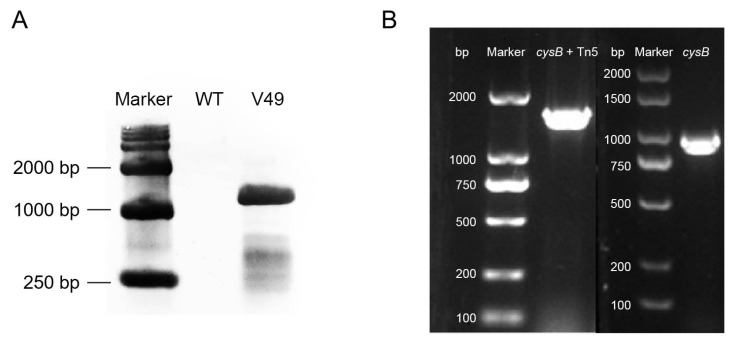
Confirmation of the Tn5 insertion of the mutant gene in the WT. (**A**) Southern blot analysis of the mutant. (**B**) Validation of the *cysB* fragment. The template for the fragment shown on the left is the genomic DNA of the mutant V49; the template for the fragment shown on the right is the genomic DNA of the WT.

**Figure 3 ijms-24-08067-f003:**
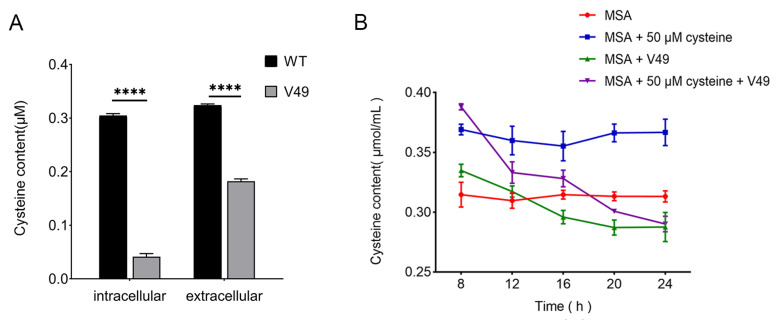
Cysteine content of the WT and mutant V49. (**A**) Intracellular and extracellular cysteine content of the WT and mutant V49. (**B**) Extracellular cysteine content of the mutant V49 at different times. ****, *p* < 0.0001.

**Figure 4 ijms-24-08067-f004:**
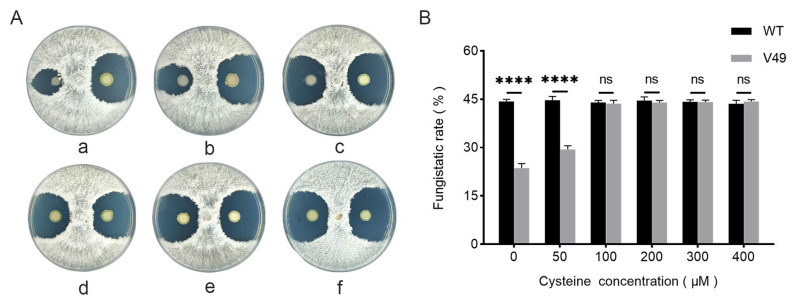
Effect of different concentrations of cysteine on the antifungal activity of the mutant V49. (**A**) Changes in the antifungal cycle of the mutant V49 after adding different concentrations of cysteine; panels (a–f) represent the addition of cysteine at concentrations of 0, 50, 100, 200, 300, and 400 μM. (**B**) Changes in the antifungal rate of the mutant V49 after the addition of different concentrations of cysteine. ns, not significant; ****, *p* < 0.0001.

**Figure 5 ijms-24-08067-f005:**
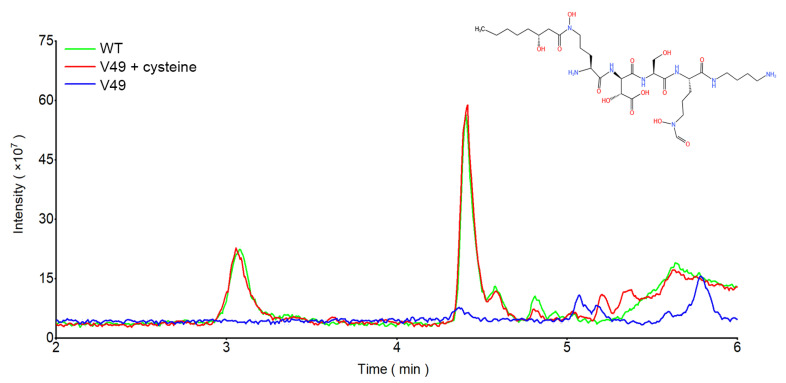
Comparison of different peaks and the corresponding substances by HPLC–MS.

**Figure 6 ijms-24-08067-f006:**
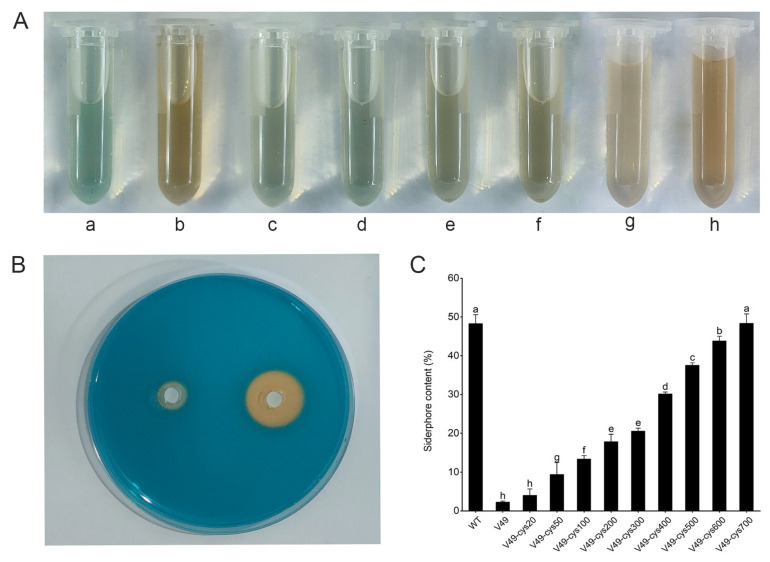
Effect of different concentrations of cysteine on the yield of siderophores. (**A**) Color development in the CAS assay solution after the addition of different concentrations of cysteine. (a) CK; (b) WT; (c) mutant V49; (d–h) cysteine added at concentrations of 50, 100, 300, 500, and 700 μM. (**B**) Difference in siderophore production between the WT and mutant V49 on CAS plates. (**C**) Quantification of siderophores in the mutant V49 strain after adding different concentrations of cysteine. Different low case letters above columns indicate statistical differences at *p* < 0.05.

**Figure 7 ijms-24-08067-f007:**
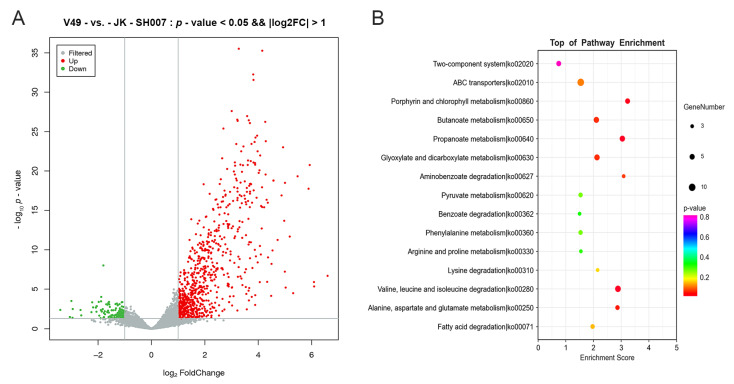
RNA-seq analysis of the WT and mutant V49. (**A**) Volcano map of DEGs. Red points denote upregulated genes; green points denote downregulated genes; gray points denote undifferentiated genes. (**B**) Top 15 enriched KEGG pathways of the DEGs. The pathway labels are on the vertical axis, and the enrichment is on the horizontal axis; the size of the dots represents the number of DEGs in the pathways, and the color of the dots represents the distinct Q-value levels.

**Figure 8 ijms-24-08067-f008:**
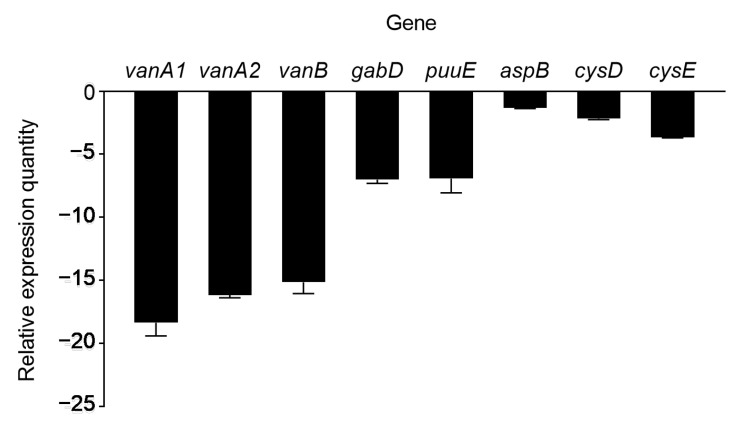
qPCR verification of related genes. The genes are *vanA1*, *vanB*, and *vanA2* for 2 iron, 2 sulfur cluster binding; *gabD* for succinate-semialdehyde dehydrogenase (NADP^+^) activity; *puuE* for 4-aminobutyrate aminotransferase activity; *aspB* for aspartate aminotransferase activity; *cysD* for cysteine desulfurase synthesis and *cysE* for serine acetyltransferase synthesis.

**Figure 9 ijms-24-08067-f009:**
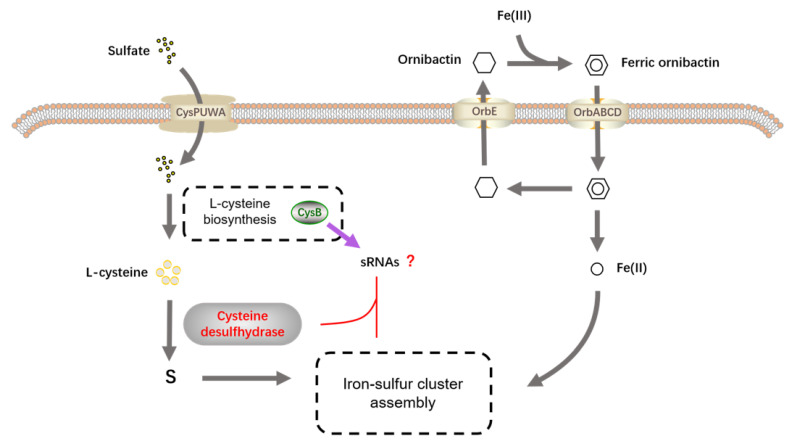
Model diagram of the possible mechanism of iron–sulfur cluster synthesis influenced by cysteine. The gray arrow indicates the involvement of cysteine and siderophore in the synthesis of iron sulfur clusters, while the purple arrow indicates possible involvement, and the “ ? ” indicates further exploration.

**Table 1 ijms-24-08067-t001:** Predicted target sites and sequences of the small RNAs identified in *B. pyrrocinia* JK-SH007.

Gene Name	Predicted Target Sites	Sequence (5′–3′)
Bp_007_sr1	Iron–sulfur cluster assembly accessory protein	GCGCGGCATGAGCGCCGCGCGCGCCGCCGCGATCTGGGCCGCGCTGGCCGCCGTGGCCGCGTTGCTGTTCGTCGCATCGCTGTCGATCGGCAGCGTGCCGATGT
Bp_007_sr2	Iron binding protein IscA for iron–sulfur cluster assembly	CTTGCCGTTCGGATCAAACAAAGAAGCGGCAAGGCGATCCCGAAGCCGCTTCATTTACCCTGGTACC
Bp_007_sr3	Ferric uptake regulation protein FUR/Iron–sulfur binding oxidoreductase	AGGCGGCCCGCCGGGCACGCGGTTCGGGCCGGACGGCCGGTACCCCGGCGGTGCGGGCTGGGCCGAGGCGGC
Bp_007_sr4	cysteine desulfurase	GGCTGCGCGGGCGAAGTTGGGCGGCGCGGCGCGCGGCGCACGCGTCGGGTGCGCAGGCGGGCCGGGCGCCAGCCGAGGCTGCCGATCATACCGGAGCACCGTGAATCCTGCCGGGCGCGGCGAACCGG

**Table 2 ijms-24-08067-t002:** The primers used in this study.

No.	Primer Name	Sequence (5′–3′)	Tm (°C)
1	Tn5-F	GCAAATTTATCCTGTGGCTG	52.4
2	Tn5-R	ACGAACCGAACAGGCTTATG	55.3
3	Tmp-F	AATTCACGAACCCAGTTGACA	54.9
4	Tmp-R	TAGGCCACACGTTCAAGTG	54.9
5	pyrG2-F	AGTCACCCTCCTCAAACTCG	56.0
6	pyrG2-R	TCGTGAAGTTGTTGGCCTTG	56.0
7	CysB-F	ATGAACCTGCACCAATTTCG	54.0
8	CysB-R	TTACAGTTCGTACGATTCGGATT	54.6
9	q-*puuE*-F	GGCGTGATGTGCGATTTCT	56.1
10	q-*puuE*-R	ACGATCTGGTAGGCGGTGT	57.8
11	q-*gabD-*F	GCACGAACCGCTTCTACG	56.4
12	q-*gabD*-R	AGCGCGTCCTCGATGTGT	59.1
13	q-*vanA1*-F	GTGCTCGTGTGCGGCTATCA	60.2
14	q-*vanA1*-R	CGGCTTCGGACGCATCGC	62.0
15	q-*vanB*-F	CGCCTGACGGCACGCACTT	63.5
16	q-*vanB*-R	CGCTTGTTCCTCCTCCGTCAGATA	60.4
17	q-*vanA2*-F	ACCTCGGCTATGTCCATCTGAA	57.6
18	q-*vanA2*-R	GGCGTGATGCCGTGGAAGC	61.8
19	q-CysD-F	GCTGTTGTTCTCGGGCGGCA	63.0
20	q-CysD-R	GCAGCGTCGTCTTGCGGTTC	61.9
21	q-CysE-F	CGGGCGGTTCCTGACGGGTA	63.4
22	q-CysE-R	GAACCGATCTTCGCACCCGC	61.2

## Data Availability

We used the known genome sequence of *B. pyrrocinia* JK_SH007 (NCBI accession: GCA_022809715.1). Additionally, the raw transcriptome data of the six samples were deposited in the SRA database (Sequence Read Archive, NCBI) with accession numbers SRR22411507, SRR22411508, SRR22411509, SRR22411504, SRR22411505, and SRR22411506.

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
