# Peer review of "CysB Is a Key Regulator of the Antifungal Activity of *Burkholderia pyrrocinia* JK-SH007"

_ijms, 2023, doi:10.3390/ijms24098067_

Round 1
Reviewer 1 Report
Dear authors,
The manuscript "CysB Is a Key Regulator of the Antifungal Activity of Burkholderia pyrrocinia JK-SH007" represents an important step towards the understanding of CysB gene function on the antagonistic mechanisms of B. pyrrocinia. This is of paramout importance in plant pathology, to develop new strategies for the control of poplar canker
The introduction is well-written and with relevant references. However, I recommend the authors to add some information on poplar canker disease. What´s the impact of this disease? Where does it occur? Moreover, there are some parts in the introduction that sounds like discussion (Lines 52-81). Please revise the introduction carefully.
The material and methods and results are also adequately described. The discussion represents important information and it discusses the main results of this work.
Nevertheless I recommend the authors to improve the conclusions. You can add some information on future work/perspectives that could be done regarding the contribution of your results for the development of new control strategies or in the use of biocontrol agents in plant pathology.
I have some minor suggestions in the attached PDF. Please ensure that all species names in the references list are adequately written, as they are not in italics!
Kind Regards

Reviewer 2 Report
1. The authors should do a complementation assay to show that there are no polar effects that account for the observed phenotype in cysB mutants.
2. Figure1: The authors should mention what media they used in their growth kinetics experiment in the legend.
3.The authors should elaborate about how they measured cysteine in both intra cellular and extracellularly in the materials and methods section.
4. Figure 4B: change the Y-axis to fold change. That will be easier for the readers to understand.
5. Did the authors saw a change in suf-operon expression in their transcriptomic experiment under different cysteine concentrations. The authors should comment on that.
6. Figure 7 A: The authors should label some of the genes that showed differentially expression in their volcano plot.
7. The authors should emphasize about how Fe-S clusters play a key role in pathogenesis in the discussion section.
8. Increase of siderophores could also lead to amino acid depletion resulting in up-regulation of stringent stress response. The authors should incorporate this in the discussion section.
Reviewer 3 Report
The manuscript entitled “CysB Is a Key Regulator of the Antifungal Activity of Burkholderia Pyrrocinia JK-SH007” is about to provide the evidence for CysB as an antifungal activity regulator.
Manuscript is well written and experiment were done very well but, to published, manuscript need some modification.
Authors need to clearly descript the “with 1‰ Tmp antibiotic”, because it was not descripted before with reason, so authors need to make sure.
Authors used mutant library to find the regulator, but author did not talk how many mutants showed similar phenotype and why authors choose V49. So, authors need to descript more clearly.
At figure 1C, how authors check the fungistatic rate? because even though V49 reduce the growth of fungi, it is not clear whether V49 reduce the growth, kill the fungi, or even fu.gi static or fungicidal is not clear. So, author need to change Figure 1C.
At figure1D, it is not clear what authors want to say with that results, because even WT, and V49 with cysteine did not show significant differences with just few explain.
If authors want to say that cysteine is important for the activity, why authors did use MSA medium? if authors use defined medium without cysteine, authors may have more clear results. For example, it is not clear why authors add the cysteine in MSA medium at figure3b because there is enough cysteine in the MSA medium already and result showed similar pattern with/ without cysteine.
Also, authors check the antagonistic ability of the mutant V49 in PDA medium, but authors did not check the concentration of PDA medium.
At figure6, Authors used MSA medium which have at least 300uM of cysteine. That suggested that less than 300uM of cysteine did not influence the siderphore synthesized. Authors need to explain that.
At line 299, authors said transcription expression but just before authors talk about the sRNA function to translation initiation. I think authors have to change to translation control or transcription control.
At references, authors need to check the style of references including dot in the journal name, and italic style and capital letter for microorganism scientific name.
Round 2
Reviewer 2 Report
Everything looks good
Reviewer 3 Report
I checked the requirement to be changed